# Changing Food Patterns during the Pandemic: Exploring the Role of Household Dynamics and Income Stabilization Strategies

**Tao Lian [1], Shamsheer ul Haq [2], Pomi Shahbaz [3,*], Lei Zhao [1], Muhammad Nadeem [4] and Babar Aziz [5]**

[1] School of International Trade and Economics, University of International Business and Economics, Beijing 100029, China

[2] Department of Economics, Division of Management and Administrative Science, University of Education, Lahore 54770, Pakistan

[3] Department of Agricultural Economic, Faculty of Agriculture, Ondokuz Mayis University, Samsun 55270, Turkey

[4] College of Economics and Management, Nanjing University of Aeronautics & Astronautics, Nanjing 210016, China

[5] Department of Economics, Government College University, Lahore 54770, Pakistan

* Correspondence: pomi1781@gmail.com

**Abstract:** COVID-19 still looms as the largest risk to the agriculture, energy, and health sectors, threatening sustainable global economic development. The literature shows that the COVID-19 pandemic can divert governments' attention away from climate change, renewable energy, and food security challenges that are necessary to address for sustainable economic growth. The COVID-19 pandemic has consistently influenced environmental behaviors, as it has primarily decreased income levels and disrupted food systems worldwide. This study examined the impacts of COVID-19 on food consumption patterns, food diversity, and income challenges and explored the factors affecting food consumption patterns during the pandemic. The data collected through an online survey from 1537 Chinese households were analyzed through a paired t-test, a mixed-design ANOVA, and a logistic regression analysis. The results revealed that the consumption of the majority of individual food commodities decreased during the COVID-19 pandemic. Among the individual food items, the consumption of pork witnessed the greatest decrease during the COVID-19 pandemic compared to the normal period. The decrease in food diversity was higher for the households whose income was affected compared to the households whose income was not affected during the COVID-19 pandemic. Furthermore, the consumption quantities of various food groups declined more for highly income-affected households than for medium and slightly affected households during the pandemic. Households that adopted a dissaving income-stabilizing strategy were 47% points more likely to maintain their food consumption patterns during the pandemic. Farmers were 17% points and 19% points less likely to suffer worsened food consumption compared to self-employed and wage workers, respectively, during the pandemic. Thus, self-production methods such as kitchen gardening can assist households to maintain and improve their consumption of food commodities during the COVID-19 pandemic.

**Keywords:** COVID-19; food consumption; sustainable agriculture; sustainable development; food diversity; food security

## 1. Introduction

In the modern world, sustainable economic growth is essential for improving human lifestyles and healthy food patterns. Sustainable household practices support energy efficiency and human food consumption patterns. The COVID-19 emergency that started in late 2019 has had a significant impact on people's lives everywhere around the world [1]. According to the prior literature, pandemic situations can cause specialists to

focus on other things besides ecology, weakening pro-environmental motivations [2]. Several studies [3–6] have found increases in environmentally friendly behaviors worldwide during the COVID-19 pandemic, including recycling and waste disposal, sustainable food consumption, and energy efficiency [7]. Food and nutrition security exist when all people, at all times, have physical, social, and economic access to sufficient, safe, and nutritious food that meets their dietary needs and food preferences for an active and healthy life [8].

Food is the most important necessity required for the sustenance of human life on earth. Food provides energy and essential nutrients for body growth, but a lack of diversity and insufficient food consumption increase the risks of diet-related diseases [9]. Food diversity is the main component of diet quality, and the consumption of diversified foods is vital to reduce the risk of diseases [10]. Food diversity is also linked to improved energy and nutrient intake in developing countries. However, shifting from monotonous to diverse food item consumption among the poor is a formidable challenge for the developing world, which is home to the majority of the undernourished people in the world [11]. Families need to consume at least 20–30 types of food items in a week for healthy body development [12]. The food consumption patterns of families are determined by their socioeconomic status [13]. Poverty, illiteracy, large family sizes, and gender inequality all contribute to poor quality and lower food consumption diversity. People with a higher social status consume more diversified food items compared to those with a lower social status [14].

Cereals are the main source of energy intake for the majority of poor people all over the world. Cereals account for 70% of the total calorie provision worldwide, while this share is only 30% in developed countries. In the poorest countries, this proportion is as high as 80% [15]. The consumption of wheat has increased more rapidly compared to rice and maize over time globally. The consumption of meat has also increased by 62%, mainly due to an increase in consumption among the families of developing countries during the last six decades. Vegetables and fruits are important sources of fiber for the human body, and at least 500 g/day of combined vegetable and fruit consumption is recommended for good health. The inadequate consumption of these food commodities is a serious problem for policymakers and nutritionists around the world, especially in developing countries. Similarly, the consumption of oils and fats also increased all over the world but the use of pulses decreased considerably during the last six decades [16]. The quality of the food consumed has been improving in terms of calories and nutritional intake due to global economic growth [17], along with rising awareness among families. COVID-19 is likely to have severe impacts on the quality and quantity of food choices made by people.

COVID-19 has had multiple direct and indirect impacts on the food consumption of the people, mainly through affecting their income and disrupting food production systems [18]. The direct impacts can include health issues associated with COVID-19, while the indirect impacts, among others, may include negative effects on the livelihood sources of the people and food market imbalances attributed to the public health measures imposed worldwide [18]. Market imbalances can be either due to demand shocks caused by income losses or supply shocks instigated by labor shortages and supply chain disruptions [19]. The COVID-19 pandemic has certainly affected the smooth functioning of food markets. In many countries, farmers have had to dump their perishable commodities and milk supplies due to significant decreases in consumer demand. The public health measures taken worldwide have affected all processes (production, transportation, distribution, and consumption) in the food supply chain [20–22]. The COVID-19 strategies particularly disrupt the supply of perishable food commodities, resulting in a shorter shelf life and high transportation costs [23,24]. This disruption in the supply chain increases the prices of these food items in local markets. Moreover, half of the world's labor force could lose their jobs due to COVID-19, especially in developing countries with huge informal sectors [25]. Consumption patterns among people are also expected to change considerably during the pandemic due to increased socioeconomic disparities. People with higher socioeconomic characteristics consume more perishable commodities (vegetables and fruits) compared to those with lower socioeconomic characteristics [26,27]. COVID-19 can also affect food

quality, and people may decrease their consumption of expensive food items and move toward cheaper food commodities that are rich in calories. Moreover, COVID-19's effects will not be distributed equally among all people. Societies, regions, countries, and people that are already poor are more vulnerable to COVID-19-induced income shocks [28].

COVID-19 is having unprecedented effects on the Chinese economy. The employment sources of informal workers are at risk, and millions may lose their livelihoods due to COVID-19 [29]. This could have disastrous implications for the consumption patterns of food commodities in terms of quality and quantity [22]. The Chinese government has adopted different strategies to reduce the negative impacts of COVID-19 on the economy through different policy initiatives such as providing tax relief for corporate and small–medium enterprises, decreasing interest rates, suspending previous credits, and providing subsidies for agriculture inputs [30,31]. Moreover, different types of social policy programs have been combined and synthesized by the government, including social insurance, social assistance, and social welfare arrangements for maintaining the food consumption patterns of the Chinese people during the pandemic [32]. Although the government took different measures to stabilize the economy and protect the livelihood sources of the people, these measures are no panacea.

Despite the emergence of the literature [33–36] related to the implications of COVID-19 on food security, there has been no study on the impacts of the pandemic on the consumption of commonly used agricultural food commodities in China. Therefore, this study will add to the growing body of literature about the implications of COVID-19 by filling this gap. The first objective of the study was to assess the changes in the consumption of all commonly used individual agricultural food items in houses during the COVID-19 pandemic compared to normal periods. The second objective of the study was to determine the implications of the COVID-19-induced income shocks on the consumption of different groups of agricultural food commodities. The third objective was to determine the factors that affect the consumption of agricultural food commodities during the COVID-19 pandemic. The fourth objective of the study was to assess the impacts of the different income-stabilizing strategies adopted by these households on their consumption of different food groups.

The economic crisis brought on by this pandemic has an impact on green habits, the consumption of sustainable foods, and energy-efficient green decisions. This study will give important information to decision-makers on how people behave in terms of their environment, including their use of sustainable food sources and the potential drivers of the ongoing economic crisis. This research study will make a significant contribution to the growing body of scientific literature and aid in our understanding, foretelling, and prevention of the potential negative effects on people's environmental aspirations and behaviors brought on by the COVID-19 crisis and other potential future economic and health crises. These crises pose problems for the global economy and environment. For health professionals and politicians who place a high priority on inadequate food and nutrition security, this study will have significant ramifications. Most significantly, this study will make it easier to handle any emergency situation that might occur again in the future.

## 2. Literature Review

### 2.1. Theoretical Background and Hypothesis Development

Gonzalez-Martinez et al. [37] described that the agriculture sector in the EU has been quite resilient during the pandemic. Pu et al. [38] found that COVID-19 has severely affected the agriculture sector. The agriculture sector plays a vital role in nourishing the gigantic global population [39]. Changes in agricultural production and the food supply chain instigated by public health measures have altered global food consumption patterns [40]. A number of studies have been conducted globally to understand the impacts of COVID-19 on food intake behaviors among households, as they vary from country to country, depending on the public health measures and severity of the pandemic.

Alhusseini and Alqahtani [41] studied the dietary patterns of households in Saudi Arabia and revealed that COVID-19 has affected the eating behaviors of the people. They further stated that the quality of food consumption was higher in normal times as compared

to COVID-19 pandemic times. Di Renzo et al. [42] compared the food patterns of Italian families before and during the COVID-19 and reported a change in dietary behavior in the study population. Similarly, Ramos-Padilla et al. [43] also explored changes in eating habits and food consumption patterns among families in Ecuador.

Jia et al. [44] conducted a study on the implications of COVID-19 on the dietary patterns of youth in China. The study noted a decline in vegetable, fruit, and meat intake while observing an increase in wheat consumption by the Chinese youth during the pandemic as compared to the normal period. A study conducted by Sidor and Rzymski [45] in Poland also found significant changes in eating behavior and consumption patterns in households. The study findings indicated that households began to consume more snacks and dairy products and reduced their intake of vegetables and fruits during the pandemic. In Pakistan, Shahbaz et al. [46] studied the impact of COVID-19 on the daily intake of perishable and non-perishable commodities and found that the intake of perishable food commodities decreased more during the pandemic than non-perishable foods due to COVID-19. Maestre et al. [47] also noted a decline in the consumption of fresh food commodities and an increase in sweets and snacks during the public health measures imposed to curb the spread in Spain. Güney and Sangü [48] also witnessed a change in the food consumption patterns of families in Turkey. The study results also indicated a decline in vegetable and fruit intake during the pandemic period compared to normal periods. Therefore, this study presents the following hypotheses.

**Hypothesis (H1):** *The consumption of individual agricultural food commodities has changed significantly during the COVID-19 period compared to normal periods.*

Septiyana et al. [49] conducted a study in Indonesia and found that high-income households consumed more diversified food compared to low-income households. Teixeira et al. [50] also described a shift in the food patterns of households in Brazil and found that the vegetable intake of low-income families significantly declined during the pandemic compared to the pre-pandemic period.

**Hypothesis (H2):** *Households whose incomes have been affected more during the COVID-19 pandemic are also expected to be affected more in terms of food consumption.*

### 2.2. Selection of Explanatory Variables

Moreover, the literature related to the factors affecting the consumption of households was thoroughly reviewed before the selection of exogenous variables for this study. Based on the review of the literature, socioeconomic factors such as gender, age, education, income, and livelihood sources were considered for the study. Gender has the potential to affect food consumption because males and females behave differently in the purchasing and consumption of food commodities [51]. Moreover, females were less likely to move outside during the pandemic due to a higher perception of risk and fear of contracting COVID-19 [52]. Older people consume more energy-diluted than energy-dense food items [53]. The supply of food items was highly disturbed and mobility to the market was prohibited, especially for older people in certain countries, which affected their food consumption patterns [54]. Janssen et al. [55] also described that younger and older peoples' consumption patterns changed during the pandemic. Shahbaz et al. [56] described that households with large family sizes were more likely to be affected due to COVID-19. Education, gender, and family size influence food consumption during the COVID-19 [57]. COVID-19 has affected many people psychologically, and stress problems increased during the pandemic [58]. Stress can alter food intake patterns [59]. Livelihood is also one of the main determinants of food consumption. Some livelihood sources were affected more compared to others. For example, millions of people lost their livelihood sources during the COVID-19 pandemic [29]. Unemployment caused by COVID-19 measures affected the daily food consumption patterns of households. Moreover, COVID-19 also affected all phases of the food supply chain, resulting in higher prices and the unavailability of food commodities for households [20]. The rise in prices

of food items during the pandemic lowered the affordability of basic food items [60,61]. Therefore, this study presents the following hypothesis.

**Hypothesis (H3):** *Socioeconomic characteristics significantly affected the food consumption patterns of households during the pandemic.*

*2.3. Conceptual Framework of the Study*

The literature discussed above is sufficient to develop a conceptual framework for this study. The conceptual framework of the study comprises four main parts. The first part covers COVID-19 and public health measures, the second part covers COVID-19 impacts (health, income, food supply chain, food demand, and price), the third part covers household characteristics, and the fourth part of the study covers the food patterns of households (Figure 1). The public health measures are strategies implemented by the government to curb the spread of the pandemic in the country. The COVID-19 and public health measures' impacts are the reported negative effects on society all over the world. For example, the pandemic has affected the livelihoods of millions of workers [62], resulting in reduced income [63], limited movement and logistics [64], disturbed food supply mechanisms [65], and increased prices of food commodities globally [66]. These multidimensional impacts of COVID-19 and the related public health measures have forced people to change their dietary patterns. Furthermore, the socioeconomic backgrounds of households have influenced their food consumption patterns during the pandemic [13].

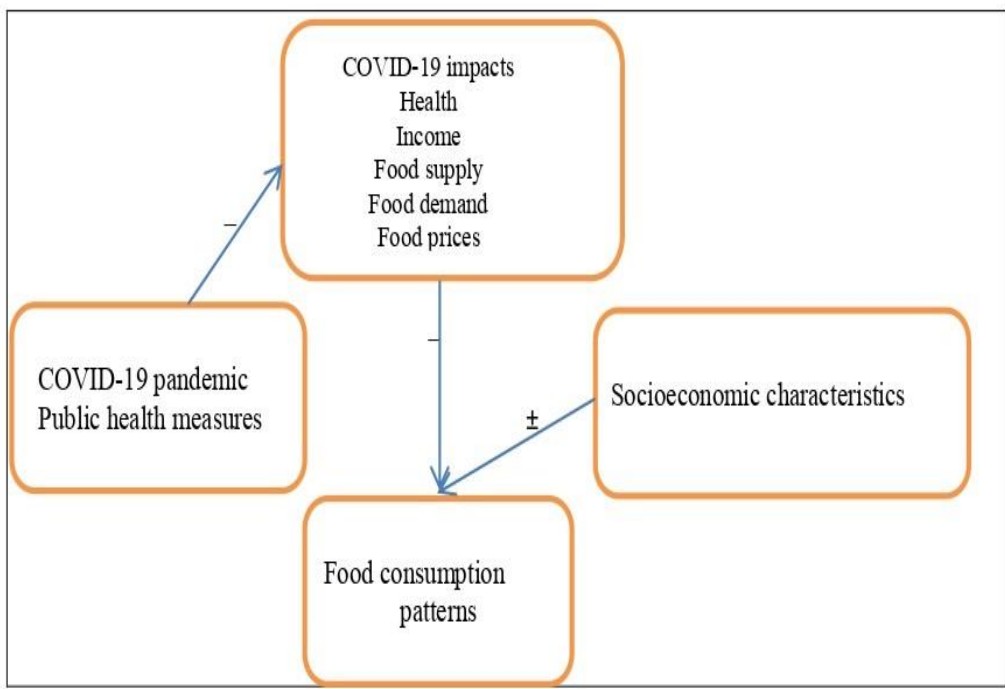

**Figure 1.** Conceptual framework of the study.

**3. Materials and Methods**

*3.1. Data Collection*

An anonymous, cross-sectional online survey using an internet questionnaire was used to collect data from 1 June to 28 June 2020. The study team used WeChat, the most widely utilized social media network in China, to promote and share the survey link within their network. The invitation to the survey was made available to all of the network members' contacts across the nation. The participants were informed that responding to the questionnaire implied their consent and that their participation was entirely voluntary. The respondents had to be Chinese citizens, at least 18 years old, and able to read and understand Chinese in order to be included in the study.

### 3.2. Questionnaire Design

The questionnaire was designed in the national language to facilitate and increase the participation and understanding of the respondents. The respondents were informed about the purpose of the study before starting the survey. The online questionnaire was divided into different sections to collect the data according to the objectives of the study. The first section was designed to gather the demographic information of the respondents, which included their age, gender, education, residential area, food market distance, marital status, and family size. The second section was devised to collect data about the consumption quantities of individual food items before and during the COVID-19 pandemic. These individual food items were grouped into cereals, vegetables, fruits, dairy products, meat and meat products, pulses, and fats and oils. Moreover, information about the consumption of different spices was also collected in the survey. The respondents' responses to variables such as affordable prices during the COVID-19 pandemic and accessibility to the food market were included in the third section of the questionnaire. Additionally, the effects of COVID-19 on the respondents' income were also explored in this section.

### 3.3. Categorizing the Families Based on COVID-19-Induced Income Effects

Families were categorized into four groups based on the COVID-19 income implications. The families who reported no effect of COVID-19 on their income were named as "unaffected families" in the study. The families experiencing less than a 34% decrease in income during the COVID-19 pandemic were categorized as "slightly affected families". Similarly, families reporting income losses ranging from 34% to 66% due to COVID-19 were included in the moderately affected group. Families whose income decreased by more than 66% in the pandemic period were classified as highly affected families in the study.

### 3.4. Measuring the Food Consumption Diversity

The consumption diversity of families was assessed using the Simpson index of diversity in this study. Higher food consumption diversity is achieved when a household consumes food commodities from all food groups in equal quantities. Therefore, to achieve higher consumption diversity, it is not only important to consume food commodities from all groups but also in equal quantities [60]. The households consuming more quantities from a single group will have lower consumption diversity compared to the households consuming uniformly from all food groups. We used the average quantities consumed by the families of each food group to calculate the consumption diversity by using the following formula:

$$CD = 1 - \sum_{g=1}^{n} p_i^2$$

where $CD$ = consumption diversity; $p_i$ = Average quantities of the $i^{\text{th}}$ group consumed; $n$ = total food groups; $g$ = 1, 2, . . . , 7

The consumption diversity values range from 0 to 1; a value near 1 describes higher consumption diversity, while a value near 0 depicts lower consumption diversity among households. The consumption diversity levels of households for both before and during the COVID-19 pandemic consumption periods were estimated in this study.

### 3.5. Empirical Analysis

A paired *t*-test was used to analyze the differences between the consumption quantities of different individual food items before and during the COVID-19 pandemic. A mixed-design analysis of variance (ANOVA) was used to examine the impacts of COVID-19-induced income shocks on the consumption of different food groups over the period (pre- and during the COVID-19 pandemic). The factors affecting the consumption of families during the COVID-19 pandemic were scrutinized through logistic regression. The general expression of the logit regression equation is given below:

$$y_i^* = \beta X_i' + \mu_i$$

where $y_i^*$ is a latent binary variable, meaning we did not observe the latent variable ($y_i^*$) directly; we observed only:

$$y_i = \begin{cases} 1 & if\ y_i^* > 0 \\ 0 & if\ y_i^* \le 0 \end{cases}$$

where $y_i$ is the binary outcome variable for families, which was developed based on the change in food consumption during the COVID-19 compared to the normal period. A family that faced worsened consumption (if the consumption difference between the normal period and COVID-19 period in mean quantities was less than zero) was assigned a value of 1, or 0 otherwise (if the consumption difference between the normal period and COVID-19 period mean quantities was equal or greater than zero). Here, $\beta_i$ represents the parameters to be estimated related to each independent variable and $\mu_i$ is the error term; *Xi* represents the socio-demographic characteristics of *i*-th household. Thus, the above equation can be rewritten as:

$$\text{pr}\ (y_i = 1 \mid X_i) = \sigma(\beta X_i') = \frac{e^{\beta X_i'}}{1 + e^{\beta X_i'}} = \frac{\exp(\beta X_i')}{1 + \exp(\beta X_i')}$$

The marginal effects were estimated to quantify and interpret the results of the explanatory variables using the following equation:

$$y_i' = \sigma(\beta X_i')\left[1 - \sigma(\beta X_i')\right]\beta_j = \frac{e^{\beta X_i'}}{\left(1 + e^{\beta X_i'}\right)2}\beta_j$$

## 4. Results and Discussion

### 4.1. Socioeconomic Characteristics of the Households

Table 1 shows the descriptive statistics of the socioeconomic characteristics of the households participating in the study. The average age of the participants was estimated to be more than 39 years. The mean schooling years of the households was more than 11 years. The education level of the households participating in this study was higher than that reported in the previous studies [44,67,68]. This was expected because the data were collected through an online survey likely to be completed by those who were using the Internet and by households that understood the questions without outside assistance. More than three-fifths of the total households belonged to urban areas. This may have been because more than 60% of the Chinese population lives in urban areas [69]. More than half of the households participating in this study were female. Similarly, the large majority of the participants were married as compared to other participants (single, widowed, etc.) in this study. Daily wages were the main source of income for more than one-third of the households participating in this research.

### 4.2. Changes in the Consumption of Individual Agricultural Commodities before and during the COVID-19

Wheat and rice are the most important staple foods for the vast majority of families in China. A significant reduction in the consumption of all cereal commodities was reported during the COVID-19 pandemic compared to normal periods (Table 2). More than half of the daily energy is provided by wheat and rice for families around the world [11]. A reduction of 550 g/month/capita in the consumption of wheat flour was noticed among individuals during the pandemic. Similarly, the families also consumed 290 g/month/capita less rice during the pandemic. The majority of families in developing countries buy cereal commodities at harvest time for the entire year. The reason for a decrease in cereal consumption may be that families might be worried about the availability of these food commodities in the next season and want to consume their present stock for a longer period of time by using a lower quantity due to the uncertainty created by the COVID-19 pandemic in food markets.

**Table 1.** Socioeconomic characteristics of the households.

| Household Characteristics | Mean | Standard Deviation |
|---|---|---|
| Age (Years) | 39.44 | 10.29 |
| Education (Years) | 11.83 | 2.05 |
| Family size (Number) | 2.90 | 0.75 |
| Distance from home to market (Km) | 5.89 | 4.76 |
| Residence area (Urban = 1 Rural = 0) | 0.61 | |
| Gender (Female = 1 0 = Male) | 0.51 | |
| Marital status (1 = Married Others = 0) | 0.63 | |
| Main livelihood source | | |
| Public employment | 0.34 | |
| Farming | 0.17 | |
| Self-employment | 0.21 | |
| Wages | 0.38 | |

Note: The standard deviations are presented only for continuous variables.

**Table 2.** Cereal consumption before and during the COVID-19 pandemic (Kg/capita/month).

| Cereals | Before COVID-19 (Mean) | During the COVID-19 Pandemic (Mean) |
|---|---|---|
| Wheat Flour | 9.12 | 8.57 * |
| Rice | 10.41 | 10.12 ** |

Note: * and ** show statistical differences at 1% and 5% respectively.

Vegetables and fruits are also part and parcel of families' daily food consumption in the country, and the results presented in Table 3 about the consumption of vegetables and fruits before and during the COVID-19 period indicate that the consumption of all vegetables and fruits reduced significantly during the COVID-19 pandemic.

Families consumed 17% fewer vegetables overall during the COVID-19 pandemic compared to the normal period. A plausible explanation may be COVID-19-induced food market shocks as well as income shocks. Vegetables are perishable commodities with a shorter life span and high transportation costs, and public health measures imposed in the country may have hindered the supply of vegetables. The other reason may have been the decrease in income due to the severe implications of COVID-19 on livelihood sources. The decrease in income may have forced families to change their dietary patterns. Kansiime et al. [70] also reported that families had to change their dietary patterns to cushion the income effects caused by COVID-19. The consumption quantity of bottle gourd presented the lowest decrease (6%) compared to other vegetables during the COVID-19 pandemic. The largest reduction was reported in the consumption of ginger (29%), followed by cabbage (26%), green pepper (25%), and garlic (24%).

Consistent with the vegetable consumption results, fruit consumption also decreased significantly during the COVID-19 pandemic compared to the normal period. Overall, the food consumption of all fruits decreased by 13–42% during the pandemic compared to the normal period. The possible reason may be inflation in the prices of fruit items instigated by COVID-19 supply chain disruptions. Grape consumption dropped by 40% during the pandemic, while peaches and strawberries dropped by 35% during the same period. Similarly, the consumption of other fruits such as apple, orange, banana, pomegranate, mango, and melon dropped by more than 24% during the pandemic period. Watermelon witnessed the lowest (13%) consumption decrease among all fruits. It was observed that the consumption of fruit items included in the "fruit" category was reduced more than any other food commodity during the pandemic. The results of this study corroborated the results found by Jia et al. [44] and Sidor and Rzymski [45], who contended that the

intake of vegetables and fruits was significantly reduced during the COVID-19 pandemic as compared to the normal period.

**Table 3.** Vegetable and fruit consumption before and during the COVID-19 pandemic (Kg/capita/month).

| Food Group/Food Item | Before COVID-19 (Mean) | During the COVID-19 Pandemic (Mean) |
|:---:|:---:|:---:|
| Vegetables | | |
| Fenugreek | 0.27 | 0.21 * |
| Sweet pepper | 0.27 | 0.23 * |
| Brinjal | 0.42 | 0.38 * |
| Cucumber | 0.64 | 0.59 ** |
| Bitter gourd | 0.45 | 0.39 ** |
| Bottle gourd | 0.45 | 0.42 *** |
| Sponge gourd | 0.47 | 0.40 * |
| Okra | 0.54 | 0.56 |
| Tinda gourd | 0.39 | 0.33 * |
| Pumpkin | 0.26 | 0.25 |
| Arum | 0.33 | 0.31 |
| Cabbage | 0.31 | 0.23 * |
| Cauliflower | 0.45 | 0.37 * |
| Carrot | 0.85 | 0.83 |
| Radish | 0.38 | 0.29 * |
| Turnip | 0.45 | 0.34 * |
| Peas | 0.52 | 0.43 * |
| Spinach | 0.56 | 0.45 * |
| Tomato | 0.77 | 0.62 * |
| Potato | 0.99 | 0.100 |
| Onion | 1.10 | 1.05 |
| Garlic | 0.28 | 0.22 * |
| Green coriander | 0.30 | 0.33 |
| Ginger | 0.19 | 0.13 ** |
| Green pepper | 0.32 | 0.24 * |
| Fruits | | |
| Orange | 1.03 | 0.79 * |
| Banana | 0.88 | 0.64 * |
| Pomegranate | 0.35 | 0.25 * |
| Mango | 1.03 | 0.74 * |
| Melon | 0.92 | 0.67 * |
| Watermelon | 1.28 | 1.11 * |
| Grapes | 0.50 | 0.29 * |
| Peach | 0.36 | 0.23 * |
| Strawberry | 0.35 | 0.22 * |
| Apple | 0.51 | 0.37 * |

Note: *, **, and *** show significant differences at 1%, 5%, and 10%, respectively.

The consumption of all dairy products except buttermilk also decreased significantly during the COVID-19 pandemic, indicating the negative effects of COVID-19 on the dietary patterns of families in the country (Table 4). The consumption of milk decreased by 7% during the COVID-19 pandemic compared to the normal period. Moreover, the consumption of butter was lowered by 18% due to the pandemic, while a decrease of 8% was noted for yogurt. These findings also indicate that the implications of COVID-19 for dairy products are less severe than for other food groups. The reason may be that most of the food items included in dairy products were prepared mostly by the families themselves.

**Table 4.** Dairy product, pulse, meat and meat product, fat and oil, and spice consumption rates before and during the COVID-19 pandemic (kg/capita/month).

| Food Group/Food Item | Before COVID-19 (Mean) | During the COVID-19 Pandemic (Mean) |
|---|---|---|
| Dairy products | | |
| Milk | 2.27 | 2.11 * |
| Butter | 0.36 | 0.30 * |
| Buttermilk | 0.91 | 0.94 |
| Yogurt | 1.01 | 0.93 * |
| Pulses | | |
| Chickpea | 0.28 | 0.24 ** |
| Split Bengal gram | 0.40 | 0.34 * |
| Green chickpea | 0.25 | 0.21 ** |
| Red beans | 0.24 | 0.19 * |
| Black gram | 0.33 | 0.28 * |
| Green lentil | 0.33 | 0.28 * |
| Brown lentil | 0.29 | 0.26 ** |
| Golden grams | 0.24 | 0.20 ** |
| Petite crimson lentils | 0.28 | 0.27 |
| Meat and meat products | | |
| Chicken | 1.27 | 1.20 * |
| Beef/Mutton | 0.39 | 0.32 * |
| Seafood | 0.97 | 0.91 * |
| Pork | 2.55 | 1.90 * |
| Fats and oil | | |
| Cooking oil | 0.83 | 0.80 * |
| Spices | | |
| Hot pepper | 0.12 | 0.08 *** |
| Salt | 0.07 | 0.06 |

Note: *, **, and *** show significant differences at 1%, 5%, and 10%, respectively.

Pulses are also an important source of energy and micronutrients for families in China. The outcome related to the consumption of pulses during the pandemic revealed that the consumption of pulses decreased considerably during the COVID-19 pandemic. The consumption rates of pulses ranged from 0.24 kg/month/capita to 0.40 kg/month/capita before the COVID-19, which were reduced significantly during the COVID-19. On average families consumed 42 g/month/capita fewer pulses during the COVID-19 pandemic compared to the normal period. COVID-19 had a significant impact on red bean consumption, which was reduced by 20% during the pandemic. Similarly, the consumption of black gram

was reduced by more than 15% during the COVID-19 pandemic. The consumption of other pulses such as chickpeas, split Bengal gram, green lentils, petite crimson lentils, and brown lentils was reduced by more than 13% during the pandemic.

The findings on the consumption of meat and meat products revealed that the consumption of these food commodities also decreased significantly during the COVID-19 pandemic. Prior to the COVID-19 pandemic, pork was the most commonly consumed item in this food group by families, and its consumption decreased by 650 g per capita per month during the pandemic. Similarly, the consumption of other meat items such as chicken and seafood was also significantly reduced during the COVID-19 pandemic.

The consumption of fats and oils was also affected by COVID-19, but this reduction was not statistically significant. The consumption of spices almost remained unchanged during the COVID-19 pandemic. Only hot pepper consumption decreased significantly during the pandemic.

The results of COVID-19 impacts indicated that the consumption of all food items was severely affected by the COVID-19 pandemic. Therefore, the hypothesis that the consumption of individual agricultural food items decreased significantly during the COVID-19 period compared to normal periods was accepted. The families also mentioned two major reasons for reducing their food consumption: (i) an increase in food prices; (ii) a decrease in their income level. Many studies conducted around the world documented increases in food prices and their impacts on food accessibility and availability. Moreover, a rise in the prices of perishable commodities such as meat, fruits, and vegetables was also reported during the pandemic worldwide [71–76]. Arndt et al. [28] reported that lockdowns have severe negative effects on the income levels of families. Therefore, the increases in the prices of food items and income losses due to COVID-19 may have forced families to change their dietary plans by reducing their consumption of food items, as the majority of food items have high income and price elasticity rates.

*4.3. Categorizing the Families Based on the Income Effects of COVID-19*

A large majority of the families' income was affected due to the COVID-19 crisis (Figure 2). Only one-fifth of the total families asserted that the pandemic had no effect on their income. More than one-third and one-fifth of the families' income levels were moderately and highly affected during the COVID-19 pandemic. These results were according to the predictions of various global organizations, which reported that the income level of the majority of people will be affected due to the pandemic.

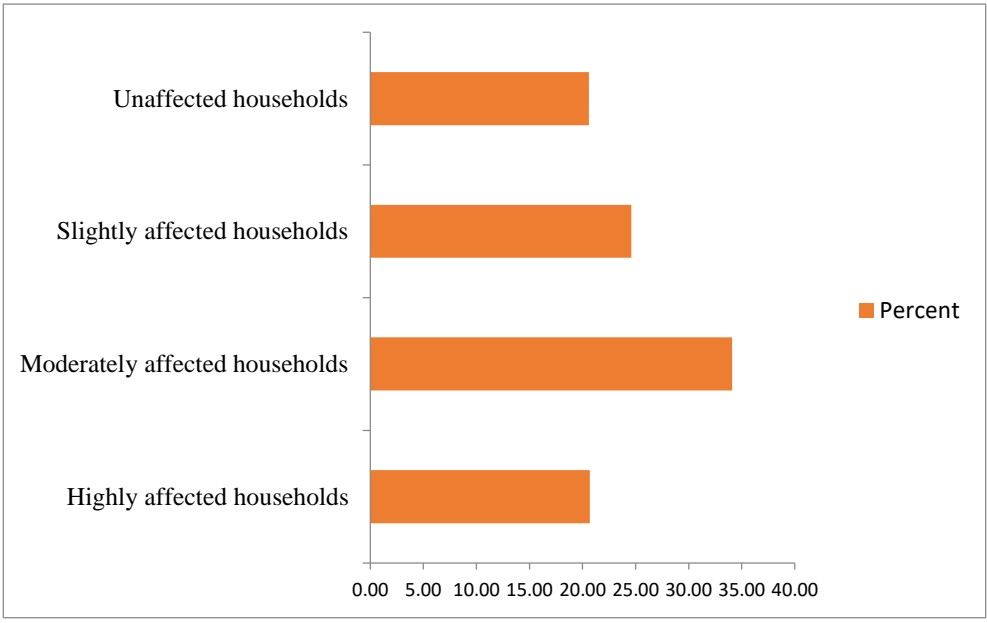

**Figure 2.** Categorizing households based on the income effects of COVID-19.

### 4.4. Implications of COVID-19 Income Effects on the Consumption of Different Agricultural Food Groups

Table 5 presents the average consumption quantities of all food groups based on the severity of the COVID-19-induced income effects on different families. The interesting outcome of the result is that despite no effect of COVID-19 on the income of the families, the consumption of vegetables, fruits, and meat and meat products decreased significantly during the pandemic, indicating that other factors such as availability, accessibility, and price increases may also be the reason for the decreased consumption of different food commodities in the country. The consumption of food groups such as cereals, meat and meat products, and oil and fats was not significantly affected by COVID-19 for those families whose income remained unaffected during the pandemic.

**Table 5.** Income effects of COVID-19 and the consumption rates of different agricultural commodity groups.

| Type of the Income Effect/Food Group | Pre COVID-19 | During the COVID-19 Pandemic |
|---|---|---|
| Highly affected families | | |
| Cereals [a] | 22.50 | 18.67 * |
| Vegetables [a] | 12.75 | 9.62 * |
| Fruits [a] | 6.39 | 4.35 * |
| Dairy products [a] | 2.37 | 1.60 * |
| Pulses [a] | 2.65 | 1.92 * |
| Meat and meat products [a] | 4.89 | 3.39 ** |
| Oil and fats [a] | 0.73 | 0.67 |
| Spices [a] | 0.94 | 0.85 * |
| Moderately affected families | | |
| Cereals [a] | 21.70 | 19.15 * |
| Vegetables [a] | 10.92 | 9.28 * |
| Fruits [a] | 6.62 | 4.63 * |
| Dairy products [a] | 2.52 | 1.87 * |
| Pulses [a] | 2.86 | 2.30 * |
| Meat and meat products [a] | 5.93 | 4.68 ** |
| Oil and fats [a] | 0.88 | 0.90 |
| Spices [a,b] | 0.89 | 0.82 * |
| Slightly affected families | | |
| Cereals [a] | 20.24 | 18.87 ** |
| Vegetables [a] | 12.68 | 11.20 * |
| Fruits [a] | 7.22 | 5.58 * |
| Dairy products [a] | 2.45 | 1.81 * |
| Pulses [a,b] | 2.94 | 2.54 * |
| Meat and meat products [a] | 5.13 | 6.10 |
| Oil and fats [a] | 1.02 | 0.99 |
| Consumption of spices [a] | 0.96 | 0.84 * |
| Unaffected families | | |
| Cereals [b] | 17.67 | 17.43 |
| Vegetables [b] | 15.63 | 14.92 ** |
| Fruits [b] | 10.79 | 10.29 ** |

**Table 5.** *Cont.*

| Type of the Income Effect/Food Group | Pre COVID-19 | During the COVID-19 Pandemic |
|:---:|:---:|:---:|
| Dairy products [b] | 3.73 | 3.36 ** |
| Pulses [b] | 3.52 | 3.17 * |
| meat and meat products [b] | 8.14 | 7.77 |
| Oil and fats [b] | 0.78 | 0.71 |
| Spices [b] | 1.00 | 0.91 * |

Superscripts [a] and [b] show significant differences among different families' income levels. Note: * and ** show significant differences at 1%, and 5%, respectively.

In the case of slightly affected families, the consumption of meat and meat products and oils and fats was not significantly affected by COVID-19. However, their consumption of cereals was significantly reduced during the COVID-19 pandemic. Similarly, the consumption of perishable commodities such as vegetables, fruits, milk, and milk products was also significantly reduced during the COVID-19 pandemic.

The moderately and highly affected families, due to COVID-19-induced income shocks, significantly reduced their consumption of every food group during the COVID-19 pandemic compared to the normal period. According to the post hoc test, all three affected family groups significantly reduced their consumption of different food groups as compared to the unaffected families. Therefore, the hypothesis that the people whose incomes were affected more during the COVID-19 pandemic were also expected to be affected more in terms of food consumption was accepted.

*4.5. Food Consumption Diversity before and during the COVID-19 Pandemic*

The income of the household plays a vital role in food accessibility, promoting both sufficient consumption and dietary diversity [77]. Diversified food consumption is associated with a strong immune system and also decreases the risk of malnutrition [78]. Table 6 presents the food consumption diversity levels of different income-affected groups before and during the COVID-19 pandemic. The slightly affected, moderately affected, and highly affected household consumption diversity rates were significantly reduced during the COVID-19 pandemic. The consumption diversity of those households whose income was unaffected by COVID-19 remained unchanged during the COVID-19 pandemic. Shahbaz et al. [56] also described that the food diversity of households significantly decreased during the pandemic compared to the normal period. The study's results also corroborate those found by Balana et al. [79], who contended that COVID-19 has severely affected the food diversity of households in Nigeria. On the other hand, a study conducted by Hirvonen et al. [80] in Somalia found no change in the dietary diversity of the households during the pandemic. The difference may be the strict public health measures imposed in China compared to Somalia, which never enforced a full lockdown to control the pandemic's spread.

**Table 6.** Consumption diversity pre- and during the COVID-19 pandemic.

| Household Categories Based on Income Effects | Before COVID-19 | During the COVID-19 Pandemic |
|:---:|:---:|:---:|
| Highly | 0.77 | 0.76 ** |
| Moderately | 0.78 | 0.77 * |
| Slightly | 0.78 | 0.76 * |
| Unaffected | 0.77 | 0.77 |

Note: * and ** show significant differences at 1% and 5%, respectively.

### 4.6. Determinants of Consumption Change during the COVID-19 Pandemic

Female-headed families were more likely to report worsened food consumption during the COVID-19 pandemic (Table 7). FAO [81] and USAID [82] also reported that females were more vulnerable to COVID-19 in comparison to males. A 1-year increase in the age of households increased the likelihood of facing worsened food consumption during the COVID-19 pandemic. One plausible explanation is that COVID-19 threatens older people more than younger people. Moreover, governments introduced special movement restrictions for older people, making it hard for them to visit the market, which may also have affected their food consumption during the pandemic. On the other hand, a 1-year increase in the education of households decreased the probability of experiencing deteriorated food consumption during the COVID-19 pandemic. The reason may have been that educated people are expected to have better knowledge of protective measures and disease transmission, enabling them to visit food markets more securely than people with lower education. Haq et al. [83] also reported the significant positive effect of education on the consumption of different food items.

**Table 7.** Determinants of consumption changes during the COVID-19 pandemic.

| Variables | Coefficients (Std. Error) | Marginal Effects |
|---|---|---|
| Constant | −5.48 * (1.109) | - |
| Gender (Female = 1) | −0.97 * (0.357) | 0.063 |
| Age (Year) | 0.10 * (0.019) | 0.006 |
| Education (Year) | −0.05 *** (0.033) | 0.004 |
| Residence area (Rural = 1) | 0.10 (0.311) | 0.007 |
| Marital status (Married = 1) | 1.43 * (0.341) | 0.094 |
| Distance from the market (Km) | 0.06 (0.048) | 0.004 |
| Family size (members) | −0.03 (0.034) | 0.002 |
| Worried about the accessibility (Yes = 1) | 0.42 (0.336) | 0.052 |
| Affordable prices during the COVID-19 (Yes = 1) | −0.66 ** (0.316) | 0.043 |
| Family reporting income effect (Yes = 1) | 1.11 * (0.193) | 0.073 |
| Livelihood source (a) | | |
| Government employment | −2.05 * (0.614) | 0.135 |
| Self-employment | 2.54 * (0.620) | 0.167 |
| Wage employment | 2.81 * (0.660) | 0.185 |

Log likelihood = −162.67: No. of Obs. = 1537: LR chi$^2$ (13) = 393.14: Prob > chi$^2$ = 0.00: Pseudo R$^2$ = 0.55. Note: *, **, and *** show significant differences at 1%, 5%, and 10%, respectively; (a) = base category farming.

Families dependent on government employment were 14% points less likely to suffer worsened food consumption during the COVID-19 pandemic compared to the families that depend on farming. This was because government employees received their salaries during the COVID-19 period just like they were receiving in the normal period. When compared to farmers, all other families whose income was based on remittances, self-employment, or wage employment were more likely to experience declining food consumption during the COVID-19 pandemic. This might have been because farming families have a chance of fulfilling their food needs by self-producing vegetables, fruits, cereals, and animals, lowering the chances of food worsened consumption during the COVID-19 pandemic. The families dependent on self-employment and wage employment were respectively 17% points and 19% points more likely to suffer worsened food consumption during the pandemic compared to farmers.

Families worried about the accessibility of food during COVID-19 pandemic were 5% points more likely to experience decreased food consumption during the pandemic than those families who were not worried about food accessibility. Arndt et al. [28] and

Chiwona-Karltun et al. [84] contended that public health measures have negatively affected economic growth and the stability of food markets. Reardon et al. [85] and Zhou and Delgado [86] stated that disruptions in the food supply chain limit the availability and accessibility of food.

The result for the affordability of food items showed that families who reported unaffordable food prices were 4% points more likely to compromise on their food consumption during the COVID-19 pandemic than those who did not report food price issues. Unaffordable prices are also linked with the purchasing power of families, which was severely affected due to the decline in income during the COVID-19 pandemic. COVID-19 had multi-dimensional effects on economies by affecting the supply and demand of goods and services. Furthermore, stringent public health measures such as mobility restrictions, associated lockdowns, and physical distancing rules implemented to contain the pandemic have resulted in increased unemployment and lower family income levels. This has ultimately affected consumer spending in the economy and the supply and demand of commodities. The families whose income levels were affected due to COVID-19-induced income shocks were 7% points more likely to face decreased food consumption than those whose income was unaffected.

### 4.7. Measuring the Effects of Income-Stabilizing Strategies on Household Consumption

Households adopted different measures to stabilize their consumption during the COVID-19 pandemic to cushion their decreased income due to the pandemic shocks. The first strategy used to minimize the impacts of COVID-19 on household food consumption was dissaving (Table 8). The results depicted that households using the dissaving strategy were 47% points more likely to maintain their food consumption than those who did not use this strategy during the COVID-19 pandemic. Households who borrowed from a bank, friends, or relatives had a 27% point greater chance of stabilizing their food consumption than those who did not borrow during the pandemic. Blundell et al. [87] also stated that saving and borrowing strategies were used by the households to mitigate the effects of COVID-19-induced shocks on their livelihoods and to maintain healthy eating patterns. Niles et al. [88] also described borrowing from friends or other families as one of the most widely adopted coping strategies to stabilize food access during the COVID-19 pandemic.

**Table 8.** The effects of income-stabilizing strategies on household consumption.

| Stabilizing Strategy | Coefficient (Std. Err.) | Marginal Effects |
| --- | --- | --- |
| Dissaving (Yes = 1) | 2.30 * (0.230) | 0.470 |
| Borrowed from family/friends/bank (Yes = 1) | 1.31 * (0.181) | 0.267 |
| Left buying non-food items (clothes, shoes, etc.) | 0.54 * (0.175) | 0.109 |
| Obtained help from Govt./NGOs working in the area (Yes = 1) | −0.31 (0.47) | −0.059 |
| Sold house items (Yes = 1) | 0.04 (0.42) | 0.008 |

Note: * shows significant difference at 1%.

The households also stopped buying other necessary items to cope with the impacts of COVID-19 on food consumption. The households who left to buy necessities (clothes, houses, etc.) other than food items were more likely to stabilize their income during the COVID-19 pandemic than those who did not leave to buy these necessities. Gupta et al. [89] and Di Crosta et al. [90] also described that individuals start to cope with different threats by purchasing a specific product, and they limit their purchasing of necessities such as food.

### 4.8. Limitations of the Study

To conclude, this study is not without limitations. First, the cross-sectional nature of the collected data does not permit the development of a causal relationship between COVID-19 and the changes in the consumption of food commodities. Other researchers can

verify the results of this by extending this research and gathering longitudinal data to develop a causal relationship between COVID-19 and the consumption of food commodities. Second, this research focused only on the consumption of commonly used agricultural food commodities; thus, the results are only generalizable to the extent that these commodities represent all agricultural food items. Future research can expand this study to all agricultural food commodities consumed in a house and can also include non-agricultural food commodities. Last, as in this survey, we relied on the recall abilities of the respondents for the consumed quantities of agricultural commodities, which may lack accuracy. Despite all of these limitations, this study provides important information on the impacts of COVID-19 on the consumption of commonly used agricultural food commodities in a house. This information can assist in the design of policies to protect people during the ongoing and future pandemics in China, as well as in other developing countries.

## 5. Conclusions and Policy Recommendations

Agriculture is pivotal to ensuring food and nutrition security during the pandemic. COVID-19 has severely affected the supply and demand of agricultural food commodities within and across the border. Thus, the COVID-19 pandemic poses some serious challenges to global food security and nutrition. This study investigated the implications of COVID-19 for the consumption of agricultural food commodities in China. Moreover, this research also explored the strategies that assist households in minimizing the impacts of the pandemic on their consumption of agricultural food commodities. Further, this study also examined the changes in average consumption quantities of all food groups based on the severity of COVID-19-induced income effects. In the end, factors affecting the food consumption of households were explored in this study.

The results regarding the consumption of individual food commodities revealed that the consumption of the vast majority of the commonly used agricultural food items was significantly reduced during the pandemic. Moreover, the findings also suggested that perishable food commodities showed greater changes than non-perishable food commodities during the COVID-19 pandemic. For example, the consumption of vegetables and fruits declined more than cereals and pulses during the pandemic. Families reduced their fruit consumption by more than 25% during the COVID-19 pandemic compared to normal periods. Wheat and rice consumption, on the other hand, decreased by 6.5% and 2.7%, respectively, as a result of COVID-19.

The findings also revealed that the consumption of all agricultural food commodities witnessed a greater decrease for those families whose income was affected more compared to the families whose income was affected less during the COVID-19 pandemic. Moreover, a decrease in the consumption of vegetables, fruits, pulses, and dairy products by unaffected income families pointed out additional challenges other than income such as the unavailability and inaccessibility of food commodities during the COVID-19 pandemic. It is recommended that direct links between farmers and consumers be established where possible and in accordance with public health safety standards. This will enhance the income of farmers and improve food and nutrition security through the availability of healthy diets for consumers during the COVID-19 pandemic.

The results also indicated a decline in the food diversity of families during the COVID-19 pandemic. Therefore, the government should create awareness among households about the benefits of consuming diversified foods. The findings also indicated that borrowing strategies were positively associated with maintaining food consumption during the pandemic. It is suggested that the government should start a loan program, especially to support families whose income has been severely affected due to the pandemic, to ensure food and nutrition security amid COVID-19.

The results of logit regression analysis showed that the male, young, and more educated-headed families were less likely to experience worsened food consumption during the COVID-19. Self-employed and wage-earning families were more likely to have worsened food consumption compared to farmers. Therefore, self-production (kitchen gar-

dening) can assist families to maintain and improve their consumption of food commodities during the COVID-19 pandemic.

**Author Contributions:** Conceptualization, T.L., P.S. and S.u.H.; methodology, S.u.H., M.N., B.A. and P.S.; validation, T.L., L.Z. and B.A.; formal analysis, S.u.H., P.S. and M.N.; investigation, T.L., L.Z. and M.N.; resources, T.L., L.Z. and B.A.; data curation, S.u.H. and M.N.; writing—original draft preparation, S.u.H., P.S. and T.L.; writing—review and editing, P.S., T.L. and M.N.; visualization, S.u.H. and L.Z.; supervision, S.u.H., P.S., T.L. and M.N.; project administration, T.L. and S.u.H. All authors have read and agreed to the published version of the manuscript.

**Funding:** This research is funded by the National Social Science Foundation of China (18AGL028).

**Institutional Review Board Statement:** The study was approved by Nanjing University of Aeronautics and Astronautics.

**Informed Consent Statement:** Informed consent was duly received.

**Data Availability Statement:** The data can be obtained from the corresponding author on reasonable request.

**Conflicts of Interest:** The authors have no relevant financial or non-financial interests to disclose.

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
