# Peer review of "Changing Food Patterns during the Pandemic: Exploring the Role of Household Dynamics and Income Stabilization Strategies"

_sustainability, doi:10.3390/su15010123_

Round 1

Reviewer 1 Report

The manuscript entitled "Sustainable Food Patterns during the pandemic: Exploring the Role of Household Dynamics and Income Stabilization Strategies" is an interesting read. I enjoyed reading your article and I think that you are addressing a timely and relevant topic. The article explores the impact of COVID-19 on dietary patterns through an online, cross-sectional survey. This study is important for maintaining sustainable dietary patterns amid continuing pandemics in the developing world. However, few revisions are required before this article can be accepted for publication. To this end, I have the following comments and suggestions for improving the quality of the paper:

1.     The Introduction section is well-written. Authors describe the issue well, provide the justification of the study and identiy the research gap through literature that is embedded in this section. However, I still think the literature review needs to be enhanced especially in a global context of sustainable food patterns. Some studies are suggested below:

https://doi.org/10.1016/j.technovation.2021.102255

https://doi.org/10.3390/ijerph18031175

https://doi.org/10.1016/j.apenergy.2021.118459

https://doi.org/10.3389/fenvs.2022.944156

2.     Please provide the explanation "**" in the footnote of Table 1.

3.     Authors should write the general form of the logit regression instead of the present one.

4.     The authors should write down the formula for figuring out the marginal effects, since the marginal effects are shown in Tables 5 and 6.

5.     Authors should also correct the interpretation of marginal effects from "%" to "% points" throughout the manuscript (Abstract, Results, and Conclusion).

6.     The authors should also remove unnecessary capitalization in Figure 1.

7.     Limitations could be written under the new subheading "Study limitations."

Author Response

First of all, we would like to thank you for taking out time from your busy schedule to comment on our manuscript. We have tried to include your every comment and suggestion in the revised manuscript. We hope that we successfully addressed your concerns. We also hope it will surely improve the quality of our research work. Please look at the attached file to check our point-by-point response to your valuable comments.

Reviewer 2 Report

Comments to Author/s

I carefully read the manuscript addressing Sustainable Food Patterns during the pandemic: Exploring the Role of Household Dynamics and Income Stabilization Strategies. The topic is interesting and worthy of research. This paper aims to examine the impacts of COVID-19 on food consumption patterns, food diversity, and income challenges. It explored the factors affecting food consumption patterns during the pandemic. The case reasonably fits the issues in the Sustainability journal's scope. I recommend this manuscript for publication after minor corrections.

To ensure that the paper suitably fits journal guidelines, I wish to offer a few suggestions to the Author/s:

First, the Author/s should include a separate section for a Literature review following the Introduction section. The literature section should be more specific and discuss the latest arguments and findings.

·       All hypotheses should be developed based on the theory and/or the literature.

·       2.4 Selection of explanatory variables section could be presented with the literature review section I proposed.

·       I believe it would be appropriate if the Author/s could more precisely locate the argument of the pandemic's effect in this section on all people's consumption instead of focusing specifically on people with lower socioeconomic characteristics.

·       The Author/s is/are highly encouraged to cite further good scientific papers published.

Second, the script lacks a solid conceptual framework based on this study. It would be highly appreciated if the Author/s could include a separate section on the theoretical framework following the manuscript's literature review section.  In addition, the Author/s may explain the theoretical impact of crisis (Health, Economics, etc.) on house food patterns, household dynamics, and income and discuss the key channels of crisis in detail. It should serve as the foundation for this study.

Third, Results and discussion: It would be much better if the Author/s could present a Descriptive statistical Analysis of the variables used.

In addition,

·       The Author/s could check and present the results of the requirements/assumptions with paired-sample t-test results as it is the parametric tool.   

·       Author/s are encouraged to properly maintain the in-text citations to show the source of information in writing. This is one of the weak points observed in this manuscript (for instance: lines 262-263). 

·       The results (For Instance: food consumption patterns ) of this study could be further improved by incorporating previous research findings to enhance academic soundness.

·       Author/s are encouraged to maintain table numbering (Table no. 4) correctly.

·       Please define ** in Table 1.

·       Author/s must correctly interpret Logistic Regression – Coefficients.

Finally, the Author/s are encouraged to correct academic writing mistakes, in-text citations, and references in the paper. Proofreading is necessary (for example, Line 420).

I hope the Author/s will address these issues and provide a better scientific contribution.

All the best!

Author Response

(The authors gave the same response as above.)

Author Response

(The authors gave the same response as above.)

Reviewer 4 Report

Recommendations:

1. Insert a comparison with the literature in the conclusion section.

2 In line 448 insert "who" before also

3 In line 451 insert "who" before had

Author Response

(The authors gave the same response as above.)

Reviewer 5 Report

The topic of the paper is interesting although it would have been even better to have it published earlier, i.e. in 2021 to inform the reader about the current implications of the pandemic. Nevertheless, this paper could become publishable material if the authors make some revisions to the current draft. Hopefully the comments below will help the authors in this regard: 

1. The text needs to be professionally proof-read and edited in order to improve the language used and remove all the typos that are present in the current draft. 

2.  The 'conclusions' section should be reworked. The text on limitations should be used to close the discussion section, while new text should be added to the concluding part to provide some recommendations for policy-makers. 

3. I suggest adding some additional references to studies that looked at the implications of COVID-19 in other countries. This could be used to better 'link' what happened in China with the Rest of the World. An example of this type of literature is the paper 'The COVID-19 pandemic and the EU agri-food sector: Member State impacts and recovery pathways' published in 'Studies in Agricultural Economics" and also the paper 'European Food Systems in a Regional Perspective: A Comparative Study of the Effect of COVID-19 on Households and City-Region Food Systems' by Jellery Millard et al. 

4. Section 3.4 should be extended (I am missing the discussion of the results)

5. The topic of food security should be also covered and further linked to the outcomes of this piece of research.  

6. Finally, and more importantly, the link with the 'sustainability' topic is totally missing. If the authors are not able to build this connection and cannot relate 'sustainability' with the dietary implications of COVID-19, I am afraid that the authors should look for an alternative outlet. 

Author Response

(The authors gave the same response as above.)

Round 2

Reviewer 3 Report

Comment: How to verify the accuracy of questionnaire results?
Authors’ Response: Dear reviewer, the results of this study requires further study with longitudinal data for verification and accuracy. We have mentioned it clearly in the limitation
section of the study.

If the author cannot verify the authenticity of the research results, the significance of the research cannot be guaranteed. It is not just mentioned in the limitation clearly.

Please find an appropriate method to verify the accuracy of the research results.

Author Response

Respected reviewer, please look at the attached file to check our response to your valuable comments.

Reviewer 5 Report

The current version of the paper is now much more informative than in its previous form. The writing style has improved, most typos have been corrected, the literature review has been extended and all my comments were addressed. 

However, there is a typo in Gonzales-Martinez. The correct spelling is Gonzalez-Martinez. The authors are advised to correct the typo and resubmit. 

Apart from that, I would like to kindly ask the authors to delete the following sentence from the manuscript: 'This will not only ensure food and nutrition 592 security but also decrease the risk of contracting COVID-19 by decreasing visits to the food market.' If no additional explanation is provided the statement could be misleading. 

My final comment is regarding Section 5. Limitations of the study. I would suggests the authors to make this section 4.8.

I am looking forward to a revised version of this manuscript which takes into account all these points. 

Author Response

(The authors gave the same response as above.)
